# LAMP-2 Is Involved in Surface Expression of RANKL of Osteoblasts In Vitro

**DOI:** 10.3390/ijms21176110

**Published:** 2020-08-25

**Authors:** Ineke D.C. Jansen, Wikky Tigchelaar-Gutter, Jolanda M. A. Hogervorst, Teun J. de Vries, Paul Saftig, Vincent Everts

**Affiliations:** 1Department of Periodontology, Academic Centre for Dentistry Amsterdam (ACTA), University of Amsterdam and Vrije Universiteit Amsterdam, 1081 LA Amsterdam, The Netherlands; teun.devries@acta.nl; 2Center for Microscopic Research, Department of Cell Biology and Histology, Academic Medical Centre, University of Amsterdam, 1105 AZ Amsterdam, The Netherlands; inimini11@gmail.com; 3Department of Oral Cell Biology, Academic Centre for Dentistry Amsterdam (ACTA), University of Amsterdam and Vrije Amsterdam, 1081 LA Amsterdam, The Netherlands; jma.hogervorst@acta.nl (J.M.A.H.); v.everts@acta.nl (V.E.); 4Biochemisches Institut, Christian-Albrechts-Universität, 24098 Kiel, Germany; psaftig@biochem.uni-kiel.de

**Keywords:** LAMP-2, RANKL, osteoblasts, osteoclasts

## Abstract

Lysosome associated membrane proteins (LAMPs) are involved in several processes, among which is fusion of lysosomes with phagosomes. For the formation of multinucleated osteoclasts, the interaction between receptor activator of nuclear kappa β (RANK) and its ligand RANKL is essential. Osteoclast precursors express RANK on their membrane and RANKL is expressed by cells of the osteoblast lineage. Recently it has been suggested that the transport of RANKL to the plasma membrane is mediated by lysosomal organelles. We wondered whether LAMP-2 might play a role in transportation of RANKL to the plasma membrane of osteoblasts. To elucidate the possible function of LAMP-2 herein and in the formation of osteoclasts, we analyzed these processes in vivo and in vitro using LAMP-2-deficient mice. We found that, in the presence of macrophage colony stimulating factor (M-CSF) and RANKL, active osteoclasts were formed using bone marrow cells from calvaria and long bone mouse bone marrow. Surprisingly, an almost complete absence of osteoclast formation was found when osteoclast precursors were co-cultured with LAMP-2 deficient osteoblasts. Fluorescence-activated cell sorting FACS analysis revealed that plasma membrane-bound RANKL was strongly decreased on LAMP-2 deficient osteoblasts. These results suggest that osteoblastic LAMP-2 is required for osteoblast-induced osteoclast formation in vitro.

## 1. Introduction

Lysosomes are defined as acidic hydrolase-rich cell organelles [1,2] and are involved in destruction or recycling of cellular and extracellular components. These components are delivered to the lysosome via phagocytic, endocytic or autophagic pathways. The single membrane of the lysosome serves as a barrier to keep the acidic environment separate from the rest of the cell’s cytoplasm. The composition of the lysosomal membrane is different from all other membranes in eukaryotic cells, due to (i) an extremely high carbohydrate content, (ii) a characteristic phospholipid composition, (iii) the presence of various ion pumps, and (iv) the presence of unique membrane proteins including the lysosome associated membrane proteins 1 and 2 (LAMP-1 and LAMP-2). These latter proteins represent about 50% of the total amount of lysosomal membrane proteins [3,4,5]. LAMPs contribute to the fusion between lysosomes and endosomes or other organelles [6]. In addition, LAMPs have been associated with inactivation of pathogens, down-regulation of surface receptors, repair of plasma membranes and loading of processed antigens onto major histocompatibility complex II (MHC class II) molecules [7,8]. LAMP-1 and LAMP-2 are both abundantly present in lysosomes, but are also expressed at low levels on the cell surface. The intensity of surface expression depends on the cell type and its maturation. LAMP-1 and LAMP-2 are also present on the cell surface of monocytes, macrophages and activated leukocytes [9,10,11].

After the generation of LAMP-1 and LAMP-2 knock-out mice it became clear that LAMP-1 and- 2 at least partially share some functions [4]. An upregulation of LAMP-2 was found in LAMP-1 deficient mice, which might explain why these mice were viable and fertile and none of the lysosomal enzyme activities appeared to be affected in these mice [12]. In contrast to a relatively mild phenotype of LAMP-1 deficient mice, LAMP-2 deficient mice showed a more severe phenotype. Huge accumulations of autophagic vacuoles were found in cells of the liver, pancreas, spleen, skeletal muscle, heart, lymph nodes, and in neutrophils. A high number of LAMP-2 deficient mice died prematurely, within 40 days after birth, probably due to an extensive accumulation of autophagic vacuoles [13,14,15]. Beertsen et al. [15] showed that mice deficient for LAMP-2 develop severe periodontitis due to an impaired phagosomal maturation of neutrophils. The fusion between lysosomes and phagosomes was hampered resulting in a defective clearance of pathogens.

A cell type with a very high number of lysosomes is the multinucleated osteoclast; it is the only cell type equipped to resorb bone. The formation of these cells depends on the interaction of receptor activator of nuclear kappa β (RANK) with its ligand (RANKL). Osteoclast precursors express RANK on their plasma membrane and RANKL is expressed by other bone cells such as osteoblasts [16] and osteocytes [17]. Honma and coworkers showed the existence of two pathways for RANKL transport to the plasma membrane: a direct transport to the membrane which is called the minor pathway and an indirect major pathway via secretory lysosomes. This latter pathway is upregulated by the binding of RANKL (expressed via the minor pathway) to RANK on pre-osteoclasts. The major pathway was suggested to play an essential role in osteoclastogenesis.

It is known that the majority of newly synthesized RANKL is stored in the Golgi apparatus and upon stimulation is transported to the plasma membrane [18,19,20]. Whether LAMP-2 is involved in RANKL transport to the membrane and subsequently osteoclast maturation is unknown.

For osteoclastic bone resorption, lysosomes, containing enzymes needed for bone matrix degradation, fuse with the membrane of the ruffled border, and secrete these enzymes in the resorption area. Since is known that LAMP1 and 2 play a role in lysosomal fusion, we wondered if they play a role in the fusion process of the lysosome with the ruffled border membrane.

Various studies have shown that bone site specific differences exist in the activity and use of various enzymes used for bone degradation by the osteoclasts of these bones [21,22,23,24]. One possible explanation for these phenotypically different osteoclasts might be related to local differences in osteoblasts, the cells that steer the formation of osteoclasts [25]. We wondered whether bone site-specific differences could be related to differences in LAMP-2 expression.

In the present study we investigated whether LAMP-2 plays a role in osteoclast activity and osteoclastogenesis by culturing bone marrow precursors from wild-type (WT) and LAMP-2 knock-out mice in the presence of M-CSF and RANKL. Next to that we assessed whether LAMP-2 has a role in RANKL transport to the membrane of osteoblasts. Therefore, bone marrow cells and osteoblasts were isolated from long bone as well as calvaria from WT and LAMP-2 knock-out mice and in a co-culture was their osteoclastic potential investigated.

## 2. Materials and Methods

### 2.1. Mice

LAMP-2 deficient mice (C57B6 × 129SV) were generated as described previously [14]. Care and use of the animals was approved by Ministerium für Energiewende, Landwirtschaft, Umwelt, Natur und Digitalisierung Schleswig-Holstein under number V242-49577/2019 at 16 September 2019.

The mice used in the different experiments were males of 7–14 weeks old. Since the LAMP-2 gene is localized on the X-chromosome, the male knock-out mice are indicated as LAMP-2-/y. In each experiment all genotypes were age-matched. The mice were sacrificed and tibiae and calvariae were isolated and used for bone marrow or osteoblast isolation or the bones were fixed for electron microscopy or micro Computed Tomography (µCT) (see below).

### 2.2. Microscopy

Tibiae and calvariae were isolated and fixed for 48 h at room temperature in 4% formaldehyde and 1% glutaraldehyde in 0.1 M sodium cacodylate buffer (pH 7.4). After removal of soft tissue from the bone samples the bones were decalcified for two weeks in 0.1 M ethylenediaminetetraacetic acid EDTA and 1% glutaraldehyde in cacodylate buffer, washed in buffer, postfixed in 1% osmium solution, washed in buffer again, dehydrated through a graded series of ethanol, and embedded in epoxy resin (LX-112). Semi-thin sections of 1 μm thickness were cut with a diamond knife, stained with methylene blue and used for (a) general morphology, (b) assessment of bone density, and (c) analysis of osteoclasts (number, and the presence of ruffled border) and osteoblasts. Ultrathin sections were made with a diamond knife, stained with uranyl and lead and examined in a Philips CM10 electron microscope (Philips Electron Optics, Eindhoven, The Netherlands).

### 2.3. Volume Density Determination of Intracellular Vesicles

For the analysis of vesicles present in the osteoblasts electron micrographs were used with a final magnification of ×6800. A point-counting method was used to quantify numbers of vesicles using a double latticed grid according to Weibel [26]. Ten osteoblasts of each bone type and genotype were micrographed and used to assess the volume density of intracellular vesicles of the osteoblasts. Data were expressed as percentage of total cytoplasm.

### 2.4. Micro Computed Tomography (µCT)

To study the effect of LAMP-2 deficiency on bone parameters, including trabecular volume and bone volume, intact tibiae and calvariae were examined from age-matched wild-type and LAMP-2 deficient mice. The bones were fixed (see under microscopy) and cleaned of soft tissue associated with the bones. The bones were scanned in a micro-computed tomography apparatus (µCT20, Scanco Medical AG, Zurich, Switzerland) as described in Giesen et al. [27].

### 2.5. Bone Marrow Cell Isolation

For each experiment, age-matched wild type and LAMP-2-/y mice were sacrificed with a peritoneal injection of sodium pentobarbital (0.1 mL Euthestate, Ceva Sante Animale, Naaldwijk, The Netherlands). Tibiae and calvariae were dissected and cleaned of soft tissue, and ground separately in a mortar with culture medium that consisted of alpha-minimal essential medium (α-MEM) (Invitrogen, Paisley, UK) supplemented with 5% fetal calf serum (HyClone, Logan, UT, USA), 100 U/mL penicillin, 100 µg/mL streptomycin and 250 ng/mL amphotericin B (Antibiotic Antimycotic solution, Sigma, St. Louis, MO, USA), and heparin (170 IE/mL; Leo Pharmaceutical Products B.V., Weesp, The Netherlands). The cell suspensions were aspirated through a 21-gauge needle and filtered over a 70 µm pore-size Cell Strainer filter (Falcon/Becton Dickinson, Franklin Lakes, NJ, USA).

### 2.6. Osteoclast Generation Using M-CSF and RANKL

Osteoclasts were generated as previously described in detail [28]. Briefly, bone marrow cells, which contain osteoclast precursors, were isolated from calvaria and long bone and washed twice in culture medium, centrifuged (5 min, 200× *g*), and plated in 96-well flat-bottom tissue-culture-treated plates (Cellstar, Greiner Bio-One, Monroe, NC) at a density of 1 × 10^5^ cells per well; alternatively, cells were seeded on bovine cortical bone slices. These bone slices of 650 µm in thickness, were made using a microslicer (Microslicer2, Metal research, Cambridge, UK).

Cells were cultured in 150 µL culture medium containing 30 ng/mL recombinant murine macrophage colony stimulating factor (M-CSF) (R&D systems, Minneapolis, MI, USA) and 20 ng/mL recombinant murine RANKL (R&D systems). Culture media were refreshed every three days. After six days of culture, wells were washed with phosphate buffered saline (PBS) and either fixed in 4% PBS buffered formaldehyde and stored at 4 °C, and used for tartrate-resistant acid phosphatase (TRAcP) staining, or dissolved in RNA lysis buffer (see below) and stored at −80 °C until RNA isolation. The wells with the bone slices were stored in milliQ water at 4 °C for the analysis of bone resorption.

### 2.7. Bone Resorption

Bone resorption was visualized after culture or co-culture of bone marrow cells (described in “osteoclast generation” or “co-culture”) on cortical bovine bone slices. In brief, the cells were removed from the bone slices with deionized water and subsequently the slices were fixed with 4% formaldehyde in 0.1 M cacodylate buffer for 60 min. This was followed by 1% osmium solution, and a dehydration in ascending concentrations of ethanol (60 to 100%), gold sputter coating (S150B sputter coater; Edwards, Crawley, UK) and then examined in a scanning electron microscope (XL20, Philips, Eindhoven, The Netherlands).

### 2.8. Isolation of Osteoblast-Like Cells

Tibiae of wild type and LAMP-2 deficient mice were cleaned of soft tissue and the bones were cut into small pieces. The fragments were incubated for 2 h at 37 °C with 2 mg/mL collagenase II (Sigma) [29]. These fragments were subsequently rinsed and placed in 25 cm^2^ culture flasks (Cellstar, Greiner Bio-One) with 5 mL Dulbecco’s modified minimal essential medium (DMEM) (Invitrogen) supplemented with 10% fetal calf serum (FCS) (HyClone) and 1% antibiotic–antimycotic solution in a humidified atmosphere of 5% CO_2_ in air at 37 °C. After a couple of days cells grew out of the bone fragments. When the cells were confluent (after 3–4 weeks) they were collected by the use of 0.25% trypsin and 0.1% EDTA (pH 7.3), and transferred to 75 cm^2^ culture flasks, designated as ‘passage one’. The cells of this passage and subsequent passages showed a typical trapezoidal shape. In order to assess their osteoblast-like nature the cells were cultured in mineralization medium, and the formation of mineralized nodules was analyzed. Since such nodules were formed, the osteoblast-like nature of the isolated cells was confirmed. For the subsequent experiments we used cells of passage 2–6.

### 2.9. Co-Culture

Osteoblast-like cells obtained from wild type and LAMP-2 deficient mice were plated in 96 well plates at a density of 8 × 10^3^ cells/well in α-MEM (Invitrogen) supplemented with 10% FCS and 1% antibiotic–antimycotic solution and 10 nM vitamin D (Sigma) in a humidified atmosphere of 5% CO_2_ in air at 37 °C. After their attachment during a 24 h pre-culture, bone marrow cells from wild type mice and LAMP-2 deficient mice were added in a concentration of 1 × 10^5^ cells per well. Cells were cultured together for 9 days in α-MEM (Invitrogen) supplemented with 10% FCS, 1% antibiotic antimycotic solution and 10 nM vitamin D.

### 2.10. Tartrate Resistant Acid Phosphatase (TRAcP) Staining

To visualize the osteoclasts a TRAcP staining was carried out using the Acid Phosphatase Leukocyte (TRAcP) Kit from Sigma according to the manufacturer’s instructions. TRAcP is an enzyme quite unique for osteoclasts and it is highly expressed by these cells. The nuclei were visualized with di-amidino-2-phenylindole-dihydrochloride (DAPI). Cells with three or more nuclei and which were TRAcP positive were considered as osteoclasts. The number of TRAcP-positive multinucleated cells on bone and on plastic was counted. The cells were grouped into one of the following categories: (i) cells with 3–5 nuclei, (ii) cells with 6–10 nuclei, and (iii) cells with more than 10 nuclei.

### 2.11. Quantitative RT-PCR

RNA from cultured bone marrow cells, osteoblasts and co-cultures was isolated using the RNeasy Mini Kit (Qiagen, Hilden, Germany) according to the manufacturer’s instructions. After measuring RNA concentration with a multilabel plate reader (Synergy HT, BioTek Instruments, Bad Friedrichshall, Germany), 100 ng RNA was reverse transcribed to cDNA for real-time quantitative PCR (QPCR) with the superscript Vilo cDNA synthesis kit (Invitrogen) or first strand synthesis kit (Fermentas, Thermo Fisher Scientific Inc., Waltham, MA, USA). The reactions were performed with the ABI PRISM 7000 (Applied Biosystems, Thermo Fisher Scientific) by using 5 ng cDNA and 300 nM of each primer in a total volume of 15 μL containing SYBR Green PCR Master Mix (SYBR Green I Dye, AmpliTaq Gold DNA polymerase, and dNTPs with dUTP instead of dTTP, Applied Biosystems), according to the manufacturer’s instructions. Samples were normalized for the expression of hypoxanthine guanine phosphoribosyl transferase (HPRT) by calculating the ΔCt (Ct gene of interest − Ct HPRT); expression of the different genes is given as 2^−(ΔCt)^ The primers used for the detection of the various genes are indicated in Table 1.

### 2.12. Immunolocalization of LAMP-2 and RANKL

The osteoclast cultures (M-CSF + RANKL cultures) were fixed after 6 days of culture and the osteoblast cultures after 10 days. For LAMP-2 localization the cells were fixed for 10 min with 4% PBS buffered formaldehyde and subsequently washed with PBS. For RANKL localization in osteoblasts 0.1% triton-X100 was added to the fixative to permeabilize the cells. Before incubation with the primary antibodies non-specific binding was blocked with “image it Fx signal enhancer” (Invitrogen/Molecular Probes, Carlsbad, CA, USA) for 30 min at ambient temperature. Primary antibodies used were rat monoclonal antibody against mouse LAMP-2 (ABL93 1:200 dilution, this antibody was developed by J.T. August, and obtained from the Developmental Studies Hybridoma Bank, created by the NICHD of the NIH, and maintained at The University of Iowa, Department of Biology, Iowa City, IA, USA) and RANKL (goat anti-mouse 1:20, R&D). Isotype IgG or non-immune polyclonal goat serum (DAKO, Glostrup, Denmark) were used as negative controls. Incubations with the first antibodies were overnight at 4 °C.

Bone slices with osteoclasts or wells with osteoblasts were washed three times with PBS and incubated for 60 min with the secondary antibody chicken-anti-rat-Alexa 488 (Invitrogen) for LAMP-2 localization and rabbit-anti-goat-Alexa 488 or rabbit-anti-goat-Alexa 647 conjugated antibody (Invitrogen) for RANKL localization. Following three PBS washes the LAMP-2 labelled osteoclast cultures were stained for F-actin using Alexa 633-phalloidin (Invitrogen) as described previously [30]. Finally, after washing, a drop of vectashield (Vector, Burlington, ON, Canada) was added to prevent quenching; the vectashield was supplemented with propidium iodide to stain nuclei. Image stacks were generated with a confocal laser scanning microscope (Leica microsystems, Wetzlar, Germany) using an argon laser (for Alexa 488 and propidium iodide) and a helium laser (for Alexa 633 and Alexa 647).

The expression of RANKL and LAMP-2 in osteoblasts was visualized with a Leica converted fluorescence microscope equipped with a digital camera (Leica IMDR DFC 320). Here the nuclei were visualized with DAPI.

### 2.13. FACS Analysis of RANKL

For FACS analysis the osteoblasts from WT and LAMP-2-/y mice of passage 4-6 were removed from 75 cm^2^ culture flasks with cell dissociation solution (Sigma C5914). Cells were labeled with a Phycoerythrin (PE) labeled rat anti-mouse RANKL or with an isotype control Rat IgG2a IgG (1:30/10^5^ osteoblasts; Biolegend, San Diego, CA, USA) for 30 min at 4 °C.

To inhibit autophagosome fusion, which might be involved in RANKL transport to the plasma membrane, bafilomycin (200 nM, Sigma) was added to osteoblast cultures one day after seeding and the culture proceeded for three days.

### 2.14. Statistics

T-tests were used to analyze isolated bone marrow cells, gene expression, bone resorption, TRAcP secretion and osteoclast formation in co-cultures. Differences were considered significant at *p* < 0.05. Data are expressed as mean values of at least 3 different experimental measurements ± SD. GraphPad Prism version 8 was used for statistical analyzes.

## 3. Results

### 3.1. LAMP-2 Is Present in Osteoclasts and Osteoblasts

First, we analyzed whether cultured osteoclasts and osteoblasts express LAMP-2. In wild-type osteoclasts LAMP-2 was mainly present in the area surrounded by actin filaments, which is considered the ruffled border area (Figure 1A). In wild type osteoblasts LAMP-2 was highly expressed (Figure 1B, left panel). The LAMP-2-/y osteoclasts and osteoblasts did not show any positive staining for LAMP-2 (osteoblasts in Figure 1B, right panel).

### 3.2. Osteoclasts Are Present in LAMP-2-/y Mice

To investigate whether LAMP-2 deficiency affects osteoclast formation and activity in vivo, osteoclasts present in calvaria and long bone sections were counted. Osteoclasts were present at similar numbers in both genotypes (Figure 2A,B).

However, they seemed less active in LAMP-2-/y mice, visible by a smaller ruffled border (Figure 2C,D) and/or not associated to the bone (Figure 2A,C).

### 3.3. Bone Volume Is Not Affected in LAMP-2-/y

In an attempt to corroborate the observation that less active osteoclasts are present in bones of the LAMP-2-/y mice, bone volume was analyzed by µCT. These measurements did not show differences in bone volume between WT and LAMP-2-/y mice (Figure 2E).

### 3.4. Osteoclasts Are Readily Formed from LAMP-2-/y Bone Marrow Precursors

To investigate a possible role of LAMP-2 in osteoclastogenesis we analyzed the formation of osteoclasts in two different ways. First, by culturing bone marrow cells from tibiae and calvariae in the presence of M-CSF and RANKL. Secondly, by co-culturing osteoclast precursors from calvaria and long bone marrow with osteoblasts.

For both types of cultures bone marrow cells were isolated from calvaria and long bone. Isolation of bone marrow cells from the calvariae revealed that those from LAMP-2-/y mice contained four times more marrow cells than wild type calvariae (Figure 3A). It was remarkable to note that the marrow of long bones from the LAMP-2-/y mice also contained significantly more bone marrow cells, but less pronounced than in calvaria (Figure 3A). This suggests that the absence of LAMP-2 affects the number of bone marrow cells. Of interest to note is that the number of isolated marrow cells of long bone and calvaria is similar for LAMP-2-/y mice, whereas normally calvaria contains far less bone marrow cells compared to long bone.

Bone marrow cells from calvaria and long bone were cultured for 6 days with M-CSF and RANKL and subsequently stained for TRAcP activity and the number of multinucleated TRAcP-positive cells was assessed (Figure 3B,C). In all cultures multinucleated osteoclasts were formed. In all the LAMP-2-/y cultures, on bone (Figure 3B) and on plastic (Figure 3C) the number of osteoclasts was higher compared to the WT cultures, but this was only statistically significant for calvaria cells cultured on bone.

The osteoclasts formed in these cultures proved to resorb bone. Scanning electron microscopy showed the presence of resorption pits in WT as well as in LAMP-2-/y bone marrow cultures (Figure 3D).

### 3.5. Vacuoles Are Abundantly Present in LAMP-2-/y Osteoblasts

It is known that a direct interaction of RANKL–RANK between osteoblasts and osteoclast precursors may steer the generation of osteoclasts [31,32]. Therefore, osteoclastogenesis was analyzed in co-cultures of osteoclast precursors and osteoblasts. Prior to the co-culture, the osteoblasts were analyzed at the ultrastructural level. Ultrathin sections of calvaria and long bones showed that osteoblasts of LAMP-2-/y mice contained a significantly higher number of vacuoles (Figure 4A,B). This phenomenon has already been observed for many tissues including liver, pancreas, spleen, kidney, skeletal and heart muscle [14,33], and now also for osteoblasts.

### 3.6. Osteoblastic LAMP-2 Is Essential for Osteoclastogenesis

To investigate the role of osteoblasts in osteoclast formation we isolated osteoblasts from wild type and LAMP-2 deficient mice and co-cultured these cells with osteoclast precursors obtained from the two mouse phenotypes.

In co-cultures with wild type osteoblasts, multinucleated osteoclasts were readily formed (Figure 4C,D). This was irrespective of the genetic background of the bone marrow cells (WT or LAMP-2-/y) and irrespective of the skeletal site the osteoblasts were isolated from. However, with osteoblasts obtained from LAMP-2-/y mice a striking effect was noted: hardly any osteoclast was formed (Figure 4C,D). This was apparent for osteoblasts obtained from both bone sites and irrespective of the origin of the precursors (WT or LAMP-2-/y). These findings indicate that expression of LAMP-2 by osteoblasts is essential for osteoblast-driven osteoclast formation.

Of interest was to note that the number of osteoclasts formed by the WT osteoblasts obtained from the two different bone sites proved to be different. In the co-cultures with WT calvaria osteoblasts, four times more osteoclasts were formed compared to the WT osteoblasts from long bone (compare Figure 4C with Figure 4D and note the difference in scale; *p* < 0.05).

### 3.7. Expression of Osteoclast-Related Genes Is Affected in the Co-Cultures

We then evaluated osteoclast and osteoblast gene expression. The data shown in the graphs are from wild type or LAMP-2-/y bone marrow cells co-cultured with osteoblasts from LAMP-2-/y or wild type mice. Osteoclasts were normally formed in co-cultures of bone marrow cells from LAMP-2-/y or WT with WT osteoblasts.

Corresponding to a decreased osteoclast formation, the mRNA expression of TRAcP, vacuolar ATPase (v-ATPase (d2 subunit) and dendritic cell-specific transmembrane protein (DC-STAMP) was lower in co-cultures with LAMP-2-/y osteoblasts and wild-type bone marrow cultures of calvaria cells, compared to co-cultures with wild-type osteoblasts (Figure 5A–C). A lower expression of TRAcP and v-ATPase was only significant for calvaria cells and not for those obtained from long bone. This is likely due to the four times lower number of osteoclasts formed in the long bone co-cultures (see Figure 4).

### 3.8. Expression of Osteoclastogenesis-Related Genes Is Affected in the Co-Cultures

No significant differences were found in RANKL expression of LAMP-2-/y co-cultures compared to the WT co-cultures (Figure 5D). A comparison between WT osteoblasts obtained from calvaria and long bone revealed a significant higher expression of RANKL in the calvaria co-cultures (Figure 5D). The expression of osteoprotegerin (OPG) was lower in co-cultures with the LAMP-2-/y osteoblasts, compared to the co-cultures with WT osteoblasts but this was only statistically significant for the calvaria cells (Figure 5E). The expression of C-X-C motif chemokine 12 (CXCL12) (or also called stromal cell derived factor-1; SDF-1), a chemokine which plays a role in migration of bone marrow cells and stimulates clustering prior to fusion [34], was significantly lower in the co-cultures with LAMP-2-/y osteoblasts (Figure 5F).

### 3.9. Lower RANKL Surface Expression of LAMP-2-/y Osteoblasts

Since osteoclasts were formed when RANKL was added to bone marrow cultures, and we hardly found osteoclast formation in the co-cultures, we considered the occurrence of a hampered RANKL plasma membrane expression by LAMP-2-/y osteoblasts. It is known that osteoblast-mediated osteoclast differentiation depends largely on RANK-RANKL interaction [32]. Ablation of RANKL, as in knockout animals, completely inhibits osteoclast formation. Thus RANK–RANKL interaction appears to be essential for osteoclast formation.

Immunolocalization of RANKL showed a strong labeling of this protein in WT as well as LAMP-2-/y osteoblasts. Not all cells, however, were labeled. About 50% of the cells were labeled whereas the others were negative. This was apparent for osteoblasts obtained from both phenotypes (Figure 6A). Since we used Triton-X100 in these localization experiments it was not possible to distinguish between plasma membrane associated label or label localized intracellularly.

To investigate membrane expression of RANKL we performed a FACS analysis. This revealed the presence of two populations, one with a high RANKL plasma membrane expression (peak at right side of WT and LAMP-2-/y figures in Figure 6B) and one with a lower plasma membrane expression (peak at left side of WT and LAMP-2-/y figures in Figure 6B). The percentage of cells expressing a high level of plasma membrane-bound RANKL was much lower for the LAMP-2-/y osteoblasts. In wild-type cultures 58% ± 17 of the cells was highly labeled; in the LAMP-2 -/y cultures this was only 23% ± 6 (Table 2). When bafilomycin was added to the cultures of osteoblasts, the plasma membrane bound label was lower in the WT as well as the LAMP-2-/y cells (Figure 6B and Table 2).

## 4. Discussion

The findings presented in this study show the involvement of LAMP-2 in osteoblast-mediated osteoclast formation. Osteoblasts isolated from mice deficient for LAMP-2 lacked the capacity to induce osteoclast formation. This effect was clearly related to the osteoblasts and not to the precursors of the osteoclasts. LAMP-2 deficient precursors not only formed osteoclasts with wild type osteoblasts but also in the presence of M-CSF and RANKL.

In an attempt to explain why LAMP-2 deficient osteoblasts were not able to induce the formation of osteoclasts, we analyzed the expression of RANKL by the osteoblasts. We found, in comparison with wild type osteoblasts, that less than half of the number of LAMP-2 deficient osteoblasts expressed RANKL on their plasma membrane, resulting in an overall significantly lower level of RANKL on LAMP-2-/y osteoblast plasma membranes. Since a similar expression of RANKL was found intracellularly, and also qPCR revealed no differences in expression, our findings show that RANKL in LAMP-2 deficient osteoblasts is not transported to the plasma membrane. The lack of membrane-associated RANKL offers a plausible explanation for the almost completely inhibited osteoclastogenesis. A hampered RANK–RANKL interaction in these co-cultures can also explain the lower expression of the osteoclast related genes TRAcP, v-ATPase and DC-STAMP. Takayanagi et al. and Yagi et al. [35,36] showed that the transcription pathway of these genes is upregulated upon RANK–RANKL binding.

How can such a low expression on the plasma membrane be explained? A lower expression of plasma membrane RANKL by osteoblasts can be seen in the context of findings presented by Hubert et al. [37]. These authors found in fibroblasts, comparable to our observations in osteoblasts, an enhanced number of (autophagic) vacuoles [37]. This was assumed to be due to a failure of fusion of autophagosomes with lysosomes. They showed that LAMP-2A has a critical role in recruitment of the co-factors synaptosomal associated protein-29 (SNAP-29) and vacuolar protein sorting-associated proteins 33a (Vps33a) which are important for the fusion of lysosomes and autophagosomes. Moreover, others have shown that most of the newly synthesized RANKL is not immediately transported to the plasma membrane, but stored in secretory lysosomes [19,38,39]. The latter authors hypothesized that two pathways exist for RANKL transport to the membrane: a direct minor pathway and an indirect major pathway via secretory lysosomes. This major pathway is supposed to be upregulated by the binding of RANKL (expressed via the minor pathway) to RANK on pre-osteoclasts [18]. The major pathway was suggested to actually regulate osteoclastogenesis. Honma and coworkers [18] suggested that RANKL and OPG form a complex in the Golgi apparatus, which is needed for proper transport and trimerization of RANKL at the plasma membrane. Kariya et al. [38] found that when the lysosomal co-factor Vps33a is knocked down, surface expression of RANKL is low. They also found that most of the newly synthesized RANKL is still present in the Golgi apparatus, indicating that, as mentioned above, the fusion between lysosomes and autophagosomes is hampered.

It is likely that the minor pathway functions normally in the LAMP-2/y osteoblasts, since LAMP-2 containing vesicles are not involved in this process and that only the major pathway is blocked in the absence of LAMP-2. In line with this assumption is the lower expression of OPG by the calvaria LAMP-2-/y osteoblasts.

It was of interest to note that the expression of CXCL12 mRNA was lower in the co-cultures with deficient osteoblasts. CXCL12 is a chemokine expressed and secreted by osteoblasts and in combination with its receptor, CXCR4, being present on bone marrow cells, it is involved in migration and survival of osteoclast precursors [40]. Next to the lower expression of RANKL, a lower membrane expression of CXCL12 might also result in the lower osteoclastogenesis potential of the co-cultures.

In spite of the fact that osteoclasts were hardly formed in vitro in the co-cultures with deficient osteoblasts, the number of osteoclasts in vivo appeared to be only a bit lower. Moreover, the LAMP-2 knock-out mice did not show a bone phenotype, thus suggesting properly functioning osteoclasts. To come up with an explanation for the differences between the in vivo and in vitro findings we assume that other cell types such as osteocytes and T-cells, from which is known that they have the capacity to express RANKL, facilitated osteoclastogenesis [41,42,43,44]. In this way the inability of osteoblasts to express RANKL might be compensated by other cell types.

Another explanation for the differences between the in vivo and in vitro findings might be related to a different expression of osteoclast-steering cytokines. It has been shown that LAMP-2-/y mice are liable to develop oral infections that cause periodontitis [15]. During infection inflammatory cytokines such as tumor necrosis factor α (TNF-α), interleukin-1 (IL-1) and/or interleukin-6 (IL-6) are enhanced. It is known that these cytokines are able to stimulate osteoclast formation by recruitment of higher numbers of osteoclast precursors. We found indeed that the bone marrow of LAMP-2-/y mice contained higher numbers of marrow cells. Such higher numbers of marrow cells may help to generate enough osteoclasts for proper bone remodeling.

To investigate whether RANKL expression is different between osteoblasts from different bone sites, we isolated osteoblasts from two types of bone: long bone and calvaria. Both types of osteoblasts were shown to depend on LAMP-2 for the generation of osteoclasts. There was, however, an intriguing difference in the number of osteoclasts formed in the co-cultures with LAMP2-/y osteoblasts compared to the co-cultures with WT osteoblasts. In line with a recent study of our group [25] osteoblasts from calvaria generated at least four times more osteoclasts than those obtained from long bone. This coincided with the expression of RANKL, being much higher by calvaria osteoblasts [25]. These findings indicate bone-site specific differences between osteoblasts in their capacity to generate osteoclasts.

The requirement of RANKL for osteoclastogenesis and osteoclast activity has been extensively investigated, however, how RANKL is transported and fuses with the plasma membrane is still not completely elucidated. Our results are the first to demonstrate a crucial role of LAMP-2 in RANKL expression at the plasma membrane of osteoblast-like cells and subsequently osteoclast formation. In conclusion, we propose that LAMP-2 is important for transportation of RANKL to the osteoblast plasma membrane, thereby facilitating osteoclastogenesis.

## Figures and Tables

**Figure 1 ijms-21-06110-f001:**
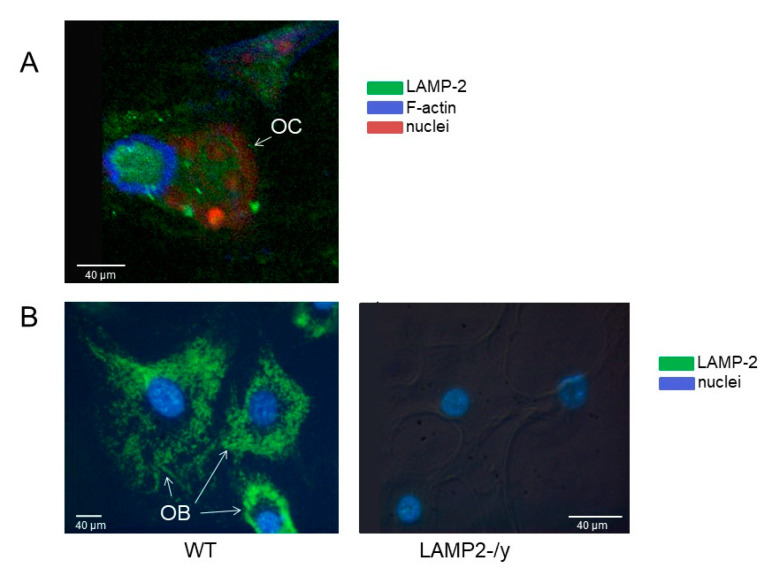
Lysosome associated membrane protein 2 (LAMP-2) localization in osteoclast and osteoblasts. LAMP-2 (visualized with alexa-488 (green), localization in (**A**) Confocal image of a wild type osteoclast (OC) on a cortical bone slice. The green LAMP-2 label is mainly present in the area surrounded by actin (visualized with phalloidin-alexa-633, in blue). This area is considered as the ruffled border area. Nuclei were stained with propidium iodide (red). (**B**) Wild-type (left) and LAMP-2-/y (right) osteoblasts (OB) cultured on plastic. Images made by a converted fluorescence microscope. The green LAMP-2 label is present throughout the cytoplasm of wild type cells, but completely absent in the LAMP-2-/y cells. Nuclei were stained with di-amidino-2-phenylindole-dihydrochloride (DAPI).

**Figure 2 ijms-21-06110-f002:**
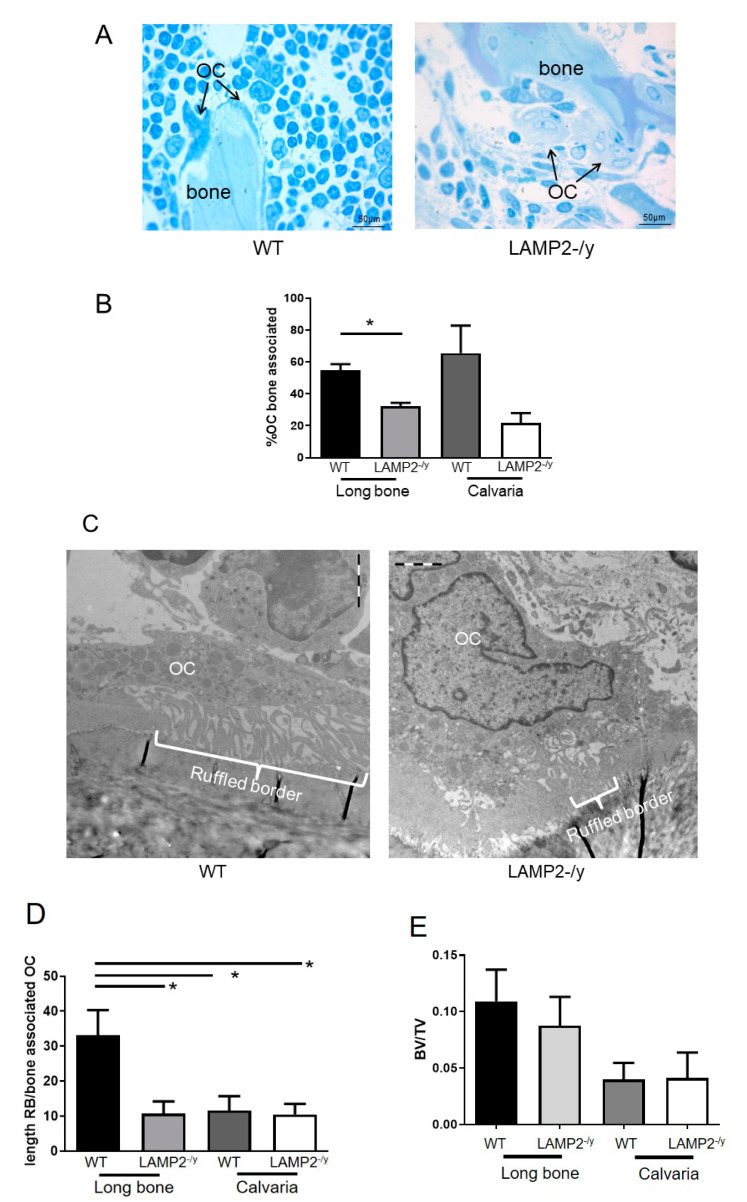
Osteoclast number and bone volume is normal in LAMP-2-/y mice. (**A**) Osteoclasts were normally present in calvariae and long bones of LAMP-2-/y mice. (**B**) Bone associated osteoclasts per total number of osteoclasts present in the light microscopic LM sections. Less bone associated osteoclasts were found in the long bone sections of LAMP-2-/y mice. (**C**) Representative electron micrographs of wild-type (WT) and LAMP-2-/y osteoclasts. In LAMP-2-/y osteoclasts the ruffled border is smaller (indicated with an accolade). (**D**) Length of the ruffled border (RB) per total bone associated cytoplasm, measured in bone associated osteoclasts (OC) (>10 sections). Long bone sections show a significant smaller ruffled border in the LAMP-2-/y mice compared to the WT. The calvaria sections also show a smaller ruffled border compared to the long bone sections, but no difference can be found between the calvaria WT and LAMP-2-/y ones. (**E**) µCT analysis showed that the bone volume/total volume (BV/TV) of LAMP-2-/y mice is equal to the WT mice of same age and gender (*n* = 6 ± SD, not significant).

**Figure 3 ijms-21-06110-f003:**
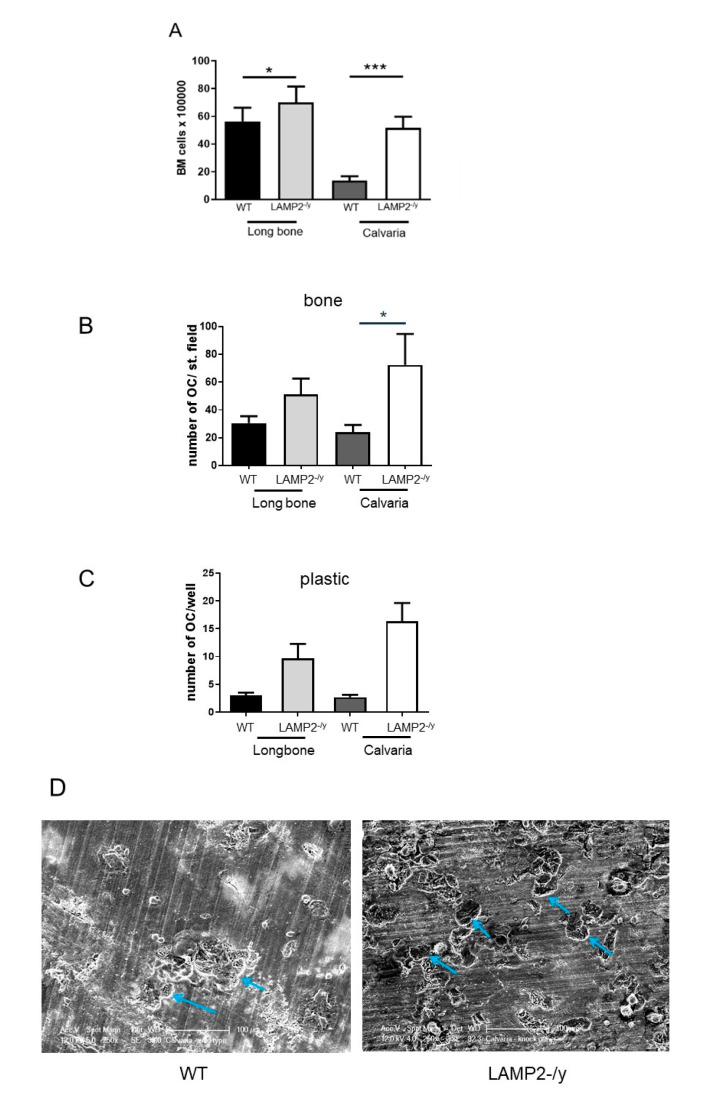
Osteoclast formation and activity in macrophage colony stimulating factor (M-CSF) and the receptor activator of nuclear kappa β ligand (RANKL) stimulated bone marrow cultures. (**A**) Number of isolated bone marrow cells from WT and LAMP-2-/y. Calvariae of LAMP-2-/y mice contained more marrow cells than the WT calvariae (*n* = 6 ± SD, *** *p* < 0.001). The number of marrow cells isolated from LAMP-2-/y long bones is slightly higher than in WT long bones. (*n* = 6 ± SD, * *p* < 0.05). Number of multinucleated cells, with more than 2 nuclei, formed after 6 days of culture with M-CSF and RANKL. Marrow cells were obtained from calvariae and long bones of wild type and LAMP-2 -/y mice. (**B**) The number of multinucleated cells generated on bone is shown. (**C**) The number of multinucleated cells generated on plastic (*n* = 6 ± SEM, * *p* < 0.05). (**D**). Scanning electron micrographs of the resorptive activity of bone marrow cells from WT and LAMP-2-/y mice cultured on cortical bone slices. In both cultures resorption pits were formed (blue arrows).

**Figure 4 ijms-21-06110-f004:**
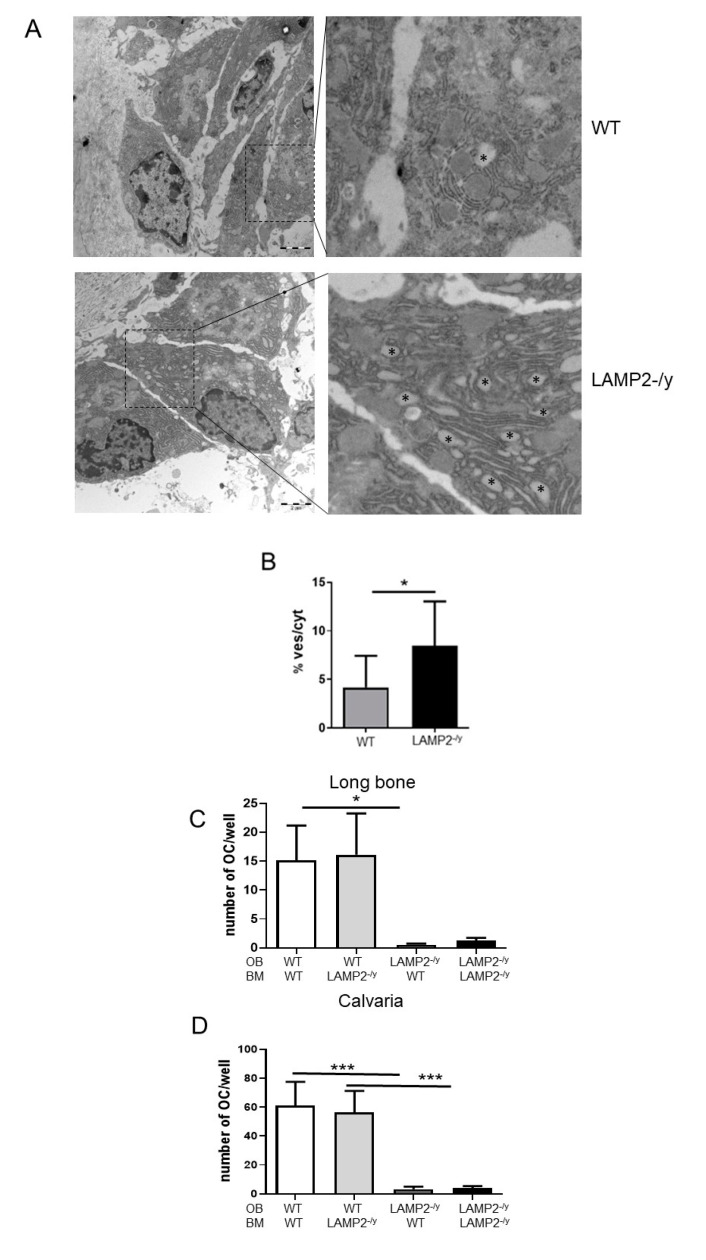
Osteoblasts of LAMP-2 deficient mice have many vacuoles and do not induce osteoclast formation. (**A**) Electron micrographs of WT and LAMP-2-/y osteoblasts. Asterisks indicate vacuoles which are much more abundant in the LAMP-2-/y osteoblasts. (**B**) The number of vacuoles per osteoblast is significantly higher in the LAMP-2-/y osteoblasts compared to WT osteoblasts (*n* = 10 ± SD, * *p* < 0.05). (**C**,**D**) Osteoblasts (OB) and bone marrow cells from long bone (**C**) and calvaria (**D**) of wild type (WT) and LAMP-2 -/y mice were co-cultured for 10 days. Wild type osteoblasts of both types of bone induced the formation of multinucleated osteoclasts. This occurred with bone marrow cells of both genotypes (WT or LAMP-2-/y). Hardly any osteoclast was generated in the co-cultures with LAMP-2-/y osteoblasts (*n* = 9, * *p* < 0.05, *** *p* < 0.001).

**Figure 5 ijms-21-06110-f005:**
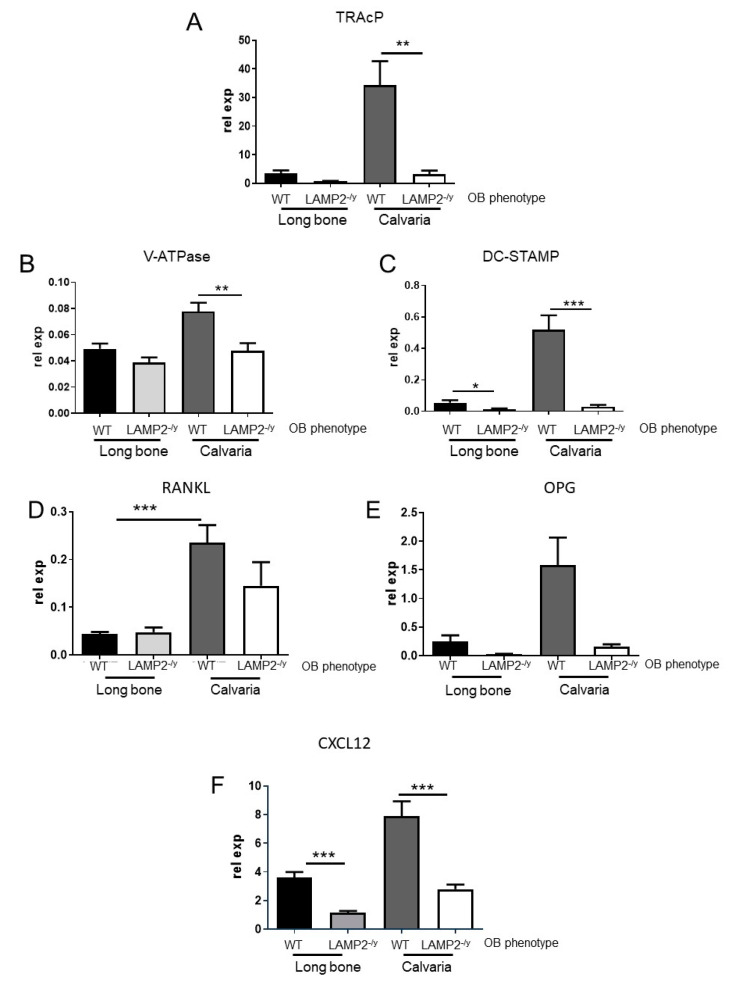
Expression of tartrate-resistant acid phosphatase (TRAcP), vacuolar- ATPase (v-ATPase), dendritic cell-specific transmembrane protein (DC-STAMP) and C-X-C motif chemokine 12 (CXCL12) is lower in co-cultures with LAMP-2 -/y osteoblasts. QPCR analysis was performed after 10 days of co-culture. Since no differences were found between the expression of bone marrow cells obtained from wild type or LAMP-2-/y mice, the data for co-cultures with wild type and LAMP-2-/y bone marrow and LAMP-2-/y were combined. Cultures with osteoblasts of LAMP-2-/y origin revealed a lower expression of (**A**) TRAcP, (**B**) v–ATPase in LAMP-2 -/y calvaria cultures and (**C**) DC-STAMP for both types of cultures (*n* = 18 ± SEM, * *p* < 0.05, ** *p* < 0.01, *** *p* < 0.001). (**D**) RANKL expression was not significantly changed in the co-cultures with LAMP-2-/y osteoblasts (*n* = 18). A significantly higher expression of RANKL was found between the calvaria and long bone co-cultures with WT osteoblasts, but not with LAMP-2-/y osteoblasts (*n* = 18 ± SEM, *** *p* < 0.0001). (**E**) Osteoprotegerin (OPG) expression is lower in the co-cultures with LAMP-2-/y osteoblasts, being only significant for calvaria (*n* = 5 ± SD, * *p* < 0.05). (**F**) CXCL12 expression is significantly lower in the co-cultures with LAMP-2-/y osteoblasts (*n* = 18 ± SEM, *** *p* < 0.0005).

**Figure 6 ijms-21-06110-f006:**
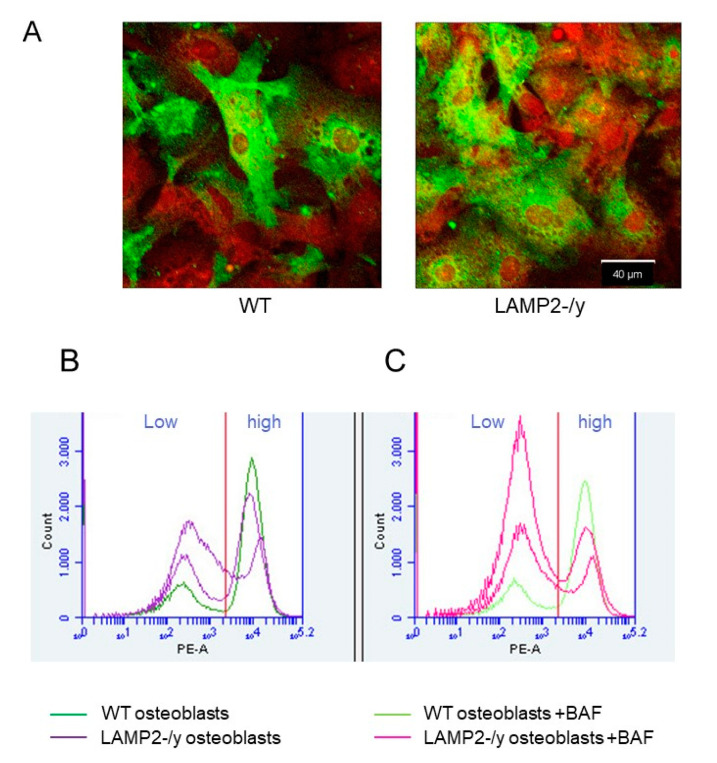
Immunolocalization and Fluorescence-activated cell sorting (FACS) analysis of RANKL in wild-type and LAMP-2 -/y osteoblasts. (**A**) WT and LAMP-2-/y osteoblasts were labeled for RANKL and visualized with alexa-488 (green), the nuclei were stained with propidium iodide (red). About 50% of the osteoblasts from both, WT and LAMP-2 -/y mice are intensely labeled for RANKL. (**B**) FACS analysis showed that membrane-bound RANKL (labeled with Phycoerythrin (PE)) was not highly expressed by all osteoblasts. Part of the osteoblasts show low membrane labelling (left side of the graph (Low)). WT osteoblasts (purple line) have more RANKL on their plasma membrane than LAMP-2-/y osteoblasts (dark blue line in graph B right side (high)). (**C**) When osteoblasts were incubated with bafilomycin, a lower number of cells were found in the high membrane labeling fraction. This is seen at the right side of the figure (high). This counts for WT as well as LAMP-2-/y cells. WT with bafilomycin (pink line), LAMP-2-/y with bafilomycin (light green line).

**Table 1 ijms-21-06110-t001:** Primers used for Quantitative PCR.

Primer	Sequence 5′–3′
HPRT	Fw: CCTAAgATgAgCgCAAgTTgAA Rv:CCACAggACTAgAACACCTgCTAA
RANKL	Fw: CTgAggCCCAgCCATTTg RV: ggAACCCgATgggATgCT
DC-STAMP	Fw: TgTATCggCTCATCTCCTCCAT Rv: gACTCCTTgggTTCCTTgCTT
v-ATPase(d2)	Fw: TggAACTAgCTCCTAACCACCT Rv: AgTTgTAAgCAgACCCTgTtgg
OPG	Fw: TCCggCgTggTgCAA Rv: ATACAgggTgCTTTCgATgAAgTC
CXCL12	FW: TgTgCATTgACCCgAAATTA RV: TCTCACATCTTgAgCCTCTTgT
TRAcP	FW: gACAAgAggTTCCAggAgACC RV: gggCTggggAAgTTCCAg

**Table 2 ijms-21-06110-t002:** Mean labeling intensity of RANKL in WT and LAMP-2-/y osteoblasts. Significantly more high-labeled cells were present in WT fraction whereas the LAMP-2-/y cell fraction contained more low-labeled cells (*n* = 8 ± SD, * *p* < 0.05).

Osteoblasts	% Low-Labeled Cells (L)	% High-Labeled Cells (R)
WT	42 ± 2	58 ± 17 *
LAMP-2-/y	77 ± 6 *	23 ± 6

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
