# Peer review of "LAMP-2 Is Involved in Surface Expression of RANKL of Osteoblasts In Vitro"

_ijms, 2020, doi:10.3390/ijms21176110_

Round 1
Reviewer 1 Report
The paper studied LAMP-2 involvement in RANKL-dependent osteoclast formation and osteoblast function in LAMP-2-deficient and wild type mice. The authors provided an evidence of the LAMP-2 importance for osteoclastogenesis due to RANKL transportation to osteoblast plasma membrane in in vitro studies. However, LAMP-2 deficiency did not affect osteoclast numbers in vivo.
Comments
- The language of the paper should be significantly improved. The authors should define in each sentence the type of cells they refer to.
- The title of the paper does not reflect the obtained results and has nothing in common with conclusions. It should be changed.
- Abstract should be rewritten as it is not clear. The language and rationality of the Abstract should be also improved. The conclusion should not contain authors assumptions.
- General: The lack of correspondence of the in vivo results to that of in vitro studies means that the in vitro model is wrong as it reflects some other process, namely, LAMP-2 involvement in RANKL-dependent osteoclast formation and osteoblast function in LAMP-2-deficient and wild type mouse cells in vitro. Therefore, the authors should describe only in vitro findings.
If the authors wish to include the results of their in vivo studies, they should present evidence for the involvement of putative “cytokines”, “osteocytes and T-cells”, and “osteoclast precursors from different bone sites” in osteoclast formation in their LAMP-2-deficient animal model. The authors should also prove the priority of the abovementioned issues over LAMP-2-deficiency in osteoclast formation process.
- All the abbreviations should be disclosed.
- Methods: The detailed description should be presented for:
- volume density determination of intracellular vesicles
- specific molecular markers of osteoblast-like cells
- the protocol for osteoclast precursor sources and preparation
- Discussion should not repeat the results and should not describe what the authors have already done and have already described in the Results section.
Author Response
Point-to-point response to the comments raised by the reviewers:
We would like to thank the reviewers for their thorough review and constructive criticisms on our manuscript entitled: LAMP-2 is involved in surface expression of RANKL of osteoblasts.
Reviewer# 1
Comments and Suggestions for Authors
The paper studied LAMP-2 involvement in RANKL-dependent osteoclast formation and osteoblast function in LAMP-2-deficient and wild type mice. The authors provided an evidence of the LAMP-2 importance for osteoclastogenesis due to RANKL transportation to osteoblast plasma membrane in in vitro studies. However, LAMP-2 deficiency did not affect osteoclast numbers in vivo.
Comments.
- The language of the paper should be significantly improved. The authors should define in each sentence the type of cells they refer to.
Answer: We included at some places, where it was unclear, the type of cells used.
- The title of the paper does not reflect the obtained results and has nothing in common with conclusions. It should be changed.
Answer: We changed the title by adding in vitro. Since the described findings were obvious only in vitro, the title will better reflect these findings.
- Abstract should be rewritten as it is not clear. The language and rationality of the Abstract should be also improved. The conclusion should not contain authors assumptions.
Answer: The author’s assumptions were removed from the abstract and we modified the abstract to make it more readable.
- General: The lack of correspondence of the in vivo results to that of in vitro studies means that the in vitro model is wrong as it reflects some other process, namely, LAMP-2 involvement in RANKL-dependent osteoclast formation and osteoblast function in LAMP-2-deficient and wild type mouse cells in vitro. Therefore, the authors should describe only in vitro findings.
If the authors wish to include the results of their in vivo studies, they should present evidence for the involvement of putative “cytokines”, “osteocytes and T-cells”, and “osteoclast precursors from different bone sites” in osteoclast formation in their LAMP-2-deficient animal model. The authors should also prove the priority of the abovementioned issues over LAMP-2-deficiency in osteoclast formation process.
Answer: We added literature confirming the involvement of osteocytes and/or T-cells in osteoclast formation in chapter 4; Discussion 6thparagraph [reference nrs 41-44].
- All the abbreviations should be disclosed.
Answer: All abbreviations are now explained in the text.
- Methods: The detailed description should be presented for:
- volume density determination of intracellular vesicles
Answer: It was included in the text, but a bit difficult to find. Now we have added it as a separate chapter under number 2.3 in M&M.
- specific molecular markers of osteoblast-like cells
Answer: We used primary cells isolated from mice bones as described by Bakker et al [29]. In order to assess their osteoblast-like nature the cells were cultured in mineralization medium, and the formation of mineralized nodules was analyzed. Since mineralized nodules were formed, the osteoblast-like nature of the isolated cells was confirmed. This has now been added to chapter 2.8 in M&M.
- the protocol for osteoclast precursor sources and preparation
Answer: In order to describe the isolation of precursors and the generation of osteoclasts in more detail two new chapters were made: Chapter 2.5 entitled: Bone marrow cell isolation and chapter 2.6 entitled: osteoclast generation.
- Discussion should not repeat the results and should not describe what the authors have already done and have already described in the Results section.
Answer: The Discussion has been modified as suggested by the reviewer. The results already described in the Results section have been removed.
Reviewer 2 Report
The manuscript aims at evaluating the role of LAMP-2 is osteoblasts-mediated osteoclastogenesis. The manuscript has a clear objective and presents several findings pointing out the role of LAMP-2 in osteoclast formation. However, some point should be clarified.
The quality of the images should be improved. The current images do not allow to assess if the authors statements match the experimental results. This is crucial as most of the evaluations are based on image analysis. Moreover, in Figure 1 there is no image showing the staining of LAMP2-/y osteoclasts. Also, Figure 1A shows the nuclei stained in red while F-actin is stained in blue. However, the nucleus of the cell seems to fill the whole cell area and looks like a F-actin staining more than nuclei staining while the F-actin staining looks more like a nucleus staining. Authors describe in the materials and methods section they stained nuclei with propidium iodade while F-actin was stained with Alexa 633-phalloidin. However, the two fluorophores used emit on really close wavelengths. Could authors explain how they were able to separate both fluorophore emission?
Moreover, there is some missing information on the materials and methods section. Authors mentioned they use bovine cortical bone slides but they did not explain how they were obtained and how long the cells were cultured on the slides before bone resorption assessment.
Also, osteoclasts counts are based just on cell morphology. Have they considered to perform immunostaining using specific osteoclast markers to facilitate osteoclast visualization?
Authors stated on section 3.7 “This is likely due to the 4 times lower number of osteoclasts formed in the long bone co-cultures (see Fig.4)”. However, gene expression was normalized by cell number. Could authors clarify this point?
Authors described in the results section: “Immunolocalization of RANKL showed a strong labeling of this protein in WT as well as LAMP-2-/y osteoblasts. Not all cells, however, were labeled. About 50% of the cells were labeled whereas the others were negative. This was apparent for osteoblasts obtained from both phenotypes (Fig. 6A)”. And, on the discussion section: “We found, in comparison with wild type osteoblasts, that less than half of the number of LAMP-2 deficient osteoblasts expressed RANKL on their plasma membrane, resulting in an overall significantly lower level of RANKL on LAMP-2-/y osteoblast plasma membranes”. Therefore, it is not clear if the expression of RANKL is indeed different between WT and LAMP-2-/y. Could authors clarify this point?
Author Response
Point-to-point response to the comments raised by the reviewers:
We would like to thank the reviewers for their thorough review and constructive criticisms on our manuscript entitled: LAMP-2 is involved in surface expression of RANKL of osteoblasts.
Reviewer#2
Comments and Suggestions for Authors
The manuscript aims at evaluating the role of LAMP-2 is osteoblasts-mediated osteoclastogenesis. The manuscript has a clear objective and presents several findings pointing out the role of LAMP-2 in osteoclast formation. However, some point should be clarified.
The quality of the images should be improved. The current images do not allow to assess if the authors statements match the experimental results. This is crucial as most of the evaluations are based on image analysis. Moreover, in Figure 1 there is no image showing the staining of LAMP2-/y osteoclasts. Also, Figure 1A shows the nuclei stained in red while F-actin is stained in blue. However, the nucleus of the cell seems to fill the whole cell area and looks like a F-actin staining more than nuclei staining while the F-actin staining looks more like a nucleus staining. Authors describe in the materials and methods section they stained nuclei with propidium iodate while F-actin was stained with Alexa 633-phalloidin. However, the two fluorophores used emit on really close wavelengths. Could authors explain how they were able to separate both fluorophore emission?
Answer: All figures have been replaced for higher quality images. We noted that the legend of Fig. 1 was missing. This is now added and also included in the text at 2.12. We made more clear that Fig. 1A is a micrograph made by confocal microscopy. The confocal microscope does not have the laser to visualize DAPI. Therefor the nuclei were stained with propidium iodide. We used the confocal microscope for this micrograph because the osteoclasts were cultured on bone slices and therefore difficult to visualize with a converted fluorescence. The osteoblasts shown in Figs 1B and C were cultured on plastic and micrographed using a converted fluorescence microscope. Here the nuclei were visualized with DAPI. This is mentioned in the last paragraph of chapter 2.12.
Moreover, there is some missing information on the materials and methods section. Authors mentioned they use bovine cortical bone slides but they did not explain how they were obtained and how long the cells were cultured on the slides before bone resorption assessment.
Answer: In section 2.6 of the Materials and Methods part we included the information on bone slices and the time of culturing. “These bone slices of 650µm in thickness, were made with a microslicer (Microslicer2, Metal research, Cambridge, UK). After six days of culture, wells were washed with PBS and……”
Also, osteoclasts counts are based just on cell morphology. Have they considered to perform immunostaining using specific osteoclast markers to facilitate osteoclast visualization?
Answer: We visualized the osteoclasts by staining them for TRAcP activity and DAPI. TRAcP is an enzyme quite unique for osteoclasts and it is highly expressed by these cells. Cells with 3 or more nuclei and being TRAcP positive were considered as osteoclasts. This extra information is included in chapter 2.10.
Authors stated on section 3.7 “This is likely due to the 4 times lower number of osteoclasts formed in the long bone co-cultures (see Fig.4)”. However, gene expression was normalized by cell number. Could authors clarify this point?
Answer: Yes indeed gene expression was normalized for the housekeeping gene as a measure for cell number. These qPCRs, however, were done for the co-cultures. In these cultures the number of osteoblasts had not changed, but the number of osteoclasts was 4 times lower. The genes that are significantly lower expressed are all of osteoclastic origin.
Authors described in the results section: “Immunolocalization of RANKL showed a strong labeling of this protein in WT as well as LAMP-2-/y osteoblasts. Not all cells, however, were labeled. About 50% of the cells were labeled whereas the others were negative. This was apparent for osteoblasts obtained from both phenotypes (Fig. 6A)”. And, on the discussion section: “We found, in comparison with wild type osteoblasts, that less than half of the number of LAMP-2 deficient osteoblasts expressed RANKL on their plasma membrane, resulting in an overall significantly lower level of RANKL on LAMP-2-/y osteoblast plasma membranes”. Therefore, it is not clear if the expression of RANKL is indeed different between WT and LAMP-2-/y. Could authors clarify this point?
Answer: We found in both WT as well as LAMP-2/y cultures that RANKL was not uniformly expressed by all cells. Some cells did not express RANKL, other cells had a low level of expression and a third category had a high level of RANKL on their plasma membrane. Next to that we found that the WT osteoblasts contained a higher number of strongly labeled cells than cultures of LAMP-2-/y cells. In the LAMP-2/y cultures many more lowly labeled cells were present (see table II). In the presence of bafilomycin, which is an inhibitor of vesicle transport, a higher number of cells show a low membrane labeling and a lower number of cells show a high membrane labeling.
Round 2
Reviewer 1 Report
Comments
- Abstract: “We found that active osteoclasts
were normally formed in macrophage colony stimulating factor (M-CSF) and RANKL driven osteoclastogenesis cultures.” The authors should specify what kind of osteoclastogenesis cultures they used.
- Introduction: The association of osteoclasts with enzymes required for bone resorption and LAMP2 deficiency is not clear. This should be clarified.
“It is known that osteoclasts present at different bone sites are different with respect to the use of enzymes (specify the enzymes) needed for bone resorption [21–24]. One possible explanation for these phenotypically different osteoclasts might be related to local differences in osteoblasts, the cells that steer the formation of osteoclasts [25]. To explore whether osteoclast precursors from different bone sites were differently affected (Please, specify how they can be differently affected) in their osteoclastic potential due to LAMP-2 deficiency, bone marrow cells and osteoblasts were isolated from long bone as well as calvaria.”
- Page 3: 2.4. MicroCT; Page 4:10. TRAcP staining. These abbreviations should be disclosed.
- Page 3: Briefly, Cells (Please, indicate, which cells were used) were washed twice in culture medium, centrifuged (5 min, 200 g), and plated in 96-well flat-bottom tissue-culturetreated plates (Cellstar, Greiner Bio-One, Monroe, NC)
- Page 4: The authors should present evidence for the purity of the isolated osteoblast-like cells.
“In order to assess their osteoblast-like nature the cells were cultured in mineralization medium, and the formation of mineralized nodules was analyzed. Since such nodules were formed, the osteoblast-like nature of the isolated cells was confirmed.”
- Page 7: The authors should assess and present data whether the difference between the number of osteoclasts per section in calvaria compared to the number of osteoclasts in long bone is significant or not. Fig.2 does not show this difference. This should be corrected.
“Note the much lower number of osteoclasts per section in calvaria compared to the number of osteoclasts in long bone. This was apparent for both genotypes, WT as well as LAMP-2-/y (Fig. 2A,B).”
- Fig 2D: The authors should disclose abbreviations used to define axeY in both graphs.
- Page 8:
“Secondly, by co-culturing osteoclast precursors (Please, indicate which osteoblast precursors were used) with osteoblasts. For both types of cultures bone marrow cells were isolated from the different bones(Please, indicate which bones were used).
After 6 days the cells of cultures (Please, indicate which cells were used) with MCSF and RANKL were stained for TRAcP activity and the number of multinucleated TRAcP-positive cells was assessed (Fig 3B,C).
- Page 8: “In all the LAMP-2-/y cultures, on bone (Fig. 3B) and on
plastic (Fig. 3C) the number of osteoclasts was higher (compared to …? This should be clarified), but this was only statistically significant for calvaria cells cultured on bone.” In addition, this statement contradicts the data presented on Fig 2 and the data stated in the Abstract. This should be corrected.
- Page 10: “To investigate the role of osteoblasts in osteoclast formation we isolated osteoblasts from wildtype and LAMP-2 deficient mice and co-cultured these cells with osteoclast precursors obtained from the two mouse phenotypes.” Protocols for osteoclast precursors generation should be clarified.
- Page 12: “The expression of osteoprotegerin (OPG) was higher (Please, specify compared to what?) in co-cultures with the LAMP-2-/y osteoblasts but this was only statistically significant for the calvaria cells”.
- Page 11: There is no reference to Table 2 in the text. This should be corrected.
- Page 11: The title of Table 2 should be written.
- Page 11: “-/y osteoblasts. Significantly more high-labeled cells were present in WT fraction whereas the LAMP-2-/y cell fraction contained more low-labeled cells (n=8 ± SD, *p<0.05).” This statement cannot be considered as a title for the Table 2. This should be corrected.
- Page 16: “In spite of the fact that osteoclasts were hardly formed in vitro in the co-cultures with deficient osteoblasts, the number of osteoclasts in vivo appeared to be unaffected.” This statement contradicts data presented on Fig 1. This should be corrected.
- Page 16: “There was, however, an intriguing difference in the formation of osteoclasts between the two osteoblast populations.” Please, specify which osteoblast populations did you mean.
- Page 17: “Our results are the first to demonstrate a crucial role for LAMP-2 in RANKL transport to the plasma membrane and subsequently osteoclast formation.” The authors cannot make any conclusions about RANKL transportation as they did not study this issue. This should be corrected.
Reviewer 2 Report
Authors have substantially modified the manuscript and therefore the work quality has been improved.
Author Response
We modified the manuscript.